# Environmental Cross-Validation of NLOS Machine Learning Classification/Mitigation with Low-Cost UWB Positioning Systems

**DOI:** 10.3390/s19245438

**Published:** 2019-12-10

**Authors:** Valentín Barral, Carlos J. Escudero, José A. García-Naya, Pedro Suárez-Casal

**Affiliations:** CITIC Research Center, Campus de Elviña, Universidade da Coruña (University of A Coruña), 15071 A Coruña, Spain; escudero@udc.es (C.J.E.); jagarcia@udc.es (J.A.G.-N.); pedro.scasal@udc.es (P.S.-C.)

**Keywords:** UWB, machine learning, neural networks, NLOS detection, indoor location algorithms

## Abstract

Indoor positioning systems based on radio frequency inherently present multipath-related phenomena. This causes ranging systems such as ultra-wideband (UWB) to lose accuracy when detecting secondary propagation paths between two devices. If a positioning algorithm uses ranging measurements without considering these phenomena, it will face critical errors in estimating the position. This work analyzes the performance obtained in a localization system when combining location algorithms with machine learning techniques applied to a previous classification and mitigation of the propagation effects. For this purpose, real-world cross-scenarios are considered, where the data extracted from low-cost UWB devices for training the algorithms come from a scenario different from that considered for the test. The experimental results reveal that machine learning (ML) techniques are suitable for detecting non-line-of-sight (NLOS) ranging values in this situation.

## 1. Introduction

As far as indoor location is concerned, ultra-wideband (UWB) technology has proven to be one of the most appropriate to achieve very accurate estimates. However, as with any radio technology, the performance of UWB-based systems is affected by typical indoor propagation phenomena such as multipath [1] yielding situations in which the transmitter and receiver are both in a non-line-of-sight (NLOS) situation. In these cases, the time of arrival (TOA) values (or time difference of arrival (TDOA) values, depending on the UWB system considered), provide data corresponding to secondary paths, which cause errors in the estimation of the actual distance that separates the transmitter from the receiver.

Several works have approached the task of detecting NLOS situations to be corrected or included in the localization process. Most of these works are based on exploiting the channel impulse response (CIR) provided by UWB hardware devices [2,3,4]. Alternatively, when it is unfeasible to obtain the CIR, the authors considered received signal strength (RSS) and ranging values measured with low-cost UWB devices as features that feed machine learning (ML) algorithms, which are employed to classify samples coming from a line-of-sight (LOS)/NLOS state and to mitigate errors in the estimated ranging values [5].

Although [5] shows the advantages and benefits of ML-based techniques for classification and mitigation in LOS and NLOS situations of UWB systems, the resulting effect on the estimation of a final position has not been studied. In addition, other works that also use other approaches based on ML usually consider the same scenario to generate the models (training, validation, and test phases) and to later evaluate them [6,7]. In this way, a part of the set of measurements is employed to build the automatic learning algorithms, whereas another part of the same set is used to evaluate the efficiency of a positioning system built over this data. This approach, although useful for finding the limits that can be reached, suffers from an important issue: It does not take into account the impact that working with measurements from a completely different physical scenario can have on the performance of algorithms.

The present work addresses this deficiency by proposing a series of experiments in which different ML models for classification and mitigation, which are trained, validated, and tested with measurements captured in a given scenario, are evaluated in a different physical scenario. Figure 1 shows the block diagram of the proposed solution, where the green blocks indicate the tasks addressed in this work. It can be seen that, unlike in the previous work [5], the blocks corresponding to the *Location Algorithm*—considering the combination of the results coming from different anchors—and *Position estimation* are new contributions of this work.

More specifically, the assessment of the proposed approach is based on the following tasks:**Measurement campaigns in two different scenarios.** Two different measurement campaigns were carried out in two distinct scenarios: The first one—referred to as the ML model-generation scenario—captured the measurements used to train, validate, and test the classification and mitigation algorithms, whereas the second one—called the evaluation scenario—compiled another different set of measurements to evaluate the performance of the previous ML algorithms over the position estimates. Both campaigns are detailed in Section 2.**Model generation.** Considering the RSS and ranging measurements recorded in the ML model-generation scenario, several ML algorithms—including multilayer Perceptrons (MLPs)—were used to train different classification and mitigation models. Section 3 details the considered algorithms, the features selected for the training, and the chosen hyper-parameters.**Positioning estimation.** The previously generated models were applied directly to the measurements obtained in the evaluation scenario. Once classified and mitigated, the resulting data were used to feed positioning algorithms—described in Section 4—to generate position estimates.**Analysis of the results.** Finally, the obtained position estimates were analyzed to assess the performance of the proposed approach. The results are shown in Section 5 together with the descriptions of the different considered configurations.

### Main Contributions

The main contributions of this article are summarized below:NLOS detection and mitigation algorithms based on ML techniques were trained considering ranging and RSS values from a UWB measurement campaign carried out in the ML model-generation scenario (see Figure 2a), whereas those models were applied to data from another UWB measurement campaign carried out in a different scenario: The evaluation scenario (see Figure 2b).Apart from some of the ML algorithms considered by the authors in previous works [5], this time another different method of this type was included in the experiments: The MLP. Thus, two different MLPs (see Section 3.1.2) were trained, validated, and tested for NLOS detection and mitigation, employing the data measured in the ML model-generation scenario.Two different location algorithms—nonlinear least squares (NLS) with a Gauss–Newton iterative procedure and an implementation of the iterative extended Kalman filter (IEKF)—were employed to fully evaluate the positioning performance of the proposed approach from the ranging data measured in the evaluation scenario (Figure 2b) after being processed with different NLOS detection and mitigation configurations (see Section 5).A comparison was made of the performance of the localization in the evaluation scenario with respect to the same scenario with an ideal situation where UWB measurements taken from the ML model-generation scenario were considered. This was possible thanks to the realistic UWB simulator that the authors developed in [8]. The results of this comparison can be seen in Section 5.2.All measurement data considered in this work are publicly available in [9,10].

## 2. Measurements

Two campaigns have been carried out in the Scientific Area Building at the University of A Coruña, Spain [11] to measure a UWB system in two different and realistic scenarios: The “ML model-generation scenario” for training, validating, and testing the algorithms, and the “evaluation scenario” for evaluating the positioning system. To this end, several UWB devices acting as anchors were placed in different known fixed locations under distinct visibility conditions, whereas the rest of the devices, acting as tags, were placed in the positions to be estimated. Notice that the ML model-generation scenario is the same as that in which the measurements that served as the basis for the works presented in [5] were obtained.

### 2.1. Hardware

Various Pozyx [12] devices were used to carry out the measurement campaigns. They are built around the DW1000 chip, a UWB transceiver manufactured by Decawave [13], and offer the following advantages: (1) Reduced cost, (2) flexibility in working with different configurations, acting as anchors and tags, with different radio parameters, and connected either to an Arduino or directly to a computer by serial port, (3) availability of additional sensors, such as an accelerometer, gyroscope, compass, and pressure sensor, which are useful for supporting localization tasks, and 4) ease of use and deployment. These advantages have converted the Pozyx solution into a great candidate to deploy UWB technology in almost any indoor scenario.

Pozyx devices rely on the high time accuracy offered by UWB to measure the signal time of flight (TOF) between transmitting and receiving antennas. To avoid any type of synchronization mechanism between devices, a protocol called two-way ranging (TWR) [14] was employed to estimate the actual signal TOF, excluding the additional signal processing time required at each communication end. More specifically, the Decawave DW1000 chip found inside Pozyx devices uses a leading edge detection (LED) mechanism to detect the first path of the incoming signal, which relies on a predefined energy threshold to detect the main path [15]. In some cases, however, this algorithm erroneously detects a path as if it were the main one. This usually happens when some obstacle blocks the direct LOS between transmitter and receiver, causing the detected signal to be a rebound from the original one; hence, the signal TOF becomes longer and, consequently, the distance is overestimated. Other error sources in these devices were detailed in [16].

Although Pozyx devices produce several measurement data types, in this work, we selected only range and RSS estimations. As explained in [5], this choice is motivated by the idea of handling as little information as possible, hence making it possible to reproduce the experiments with different—and possibly more limited—hardware. This reason, together with some particularities of the Pozyx devices [5], led us to discard the CIR, in spite of its being one of the most used techniques in the literature for the detection of NLOS [2,3,4,6].

### 2.2. Scenarios

The measurement campaigns were carried out in two different scenarios in the Scientific Area building, located on the Elviña Campus, at the University of A Coruña, Spain. In the first scenario (denoted as the “ML model-generation scenario”), measurement data were recorded to train, validate, and test the classification and mitigation algorithms, whereas in the second scenario (referred to as the “evaluation scenario”), measurement data were recorded to assess the performance of these algorithms, combined with the positioning algorithms described in Section 4, under conditions different from those experienced during the model’s generation.

Figure 2 shows the floor map of the two considered scenarios. In the ML model-generation scenario (Figure 2a), anchors were placed so that different LOS/NLOS situations could be obtained and properly labeled with respect to the positions of the tags along the corridor. In particular, the devices *A* and *B* were placed as anchors in two fixed positions, whereas a third device *C* was used as a tag to measure the respective distances at the different measurement points. The measurement points were arranged along a line with a separation of 0.2 m between them, starting from 3 m to 15 m (the hardware setup is described in [5], Section 2.4). Because of the configuration of the corridor, there was LOS propagation between the tag and anchor *B*, and NLOS propagation between the tag and anchor *A*. Figure 2a also shows the position of a third device *M*, which was connected to the computer that recorded the measurements, and its mission was to initiate a remote TWR process between each of the anchors and the tag [17]. The results, together with the RSS estimates, were sent by the serial port from the device *M* to the computer, where they were stored. In this scenario, measurements were taken for 300 s at each point with a sampling rate of 25 Hz, leading to 7500 measurements from each position.

The evaluation scenario shown in Figure 2b reproduces a typical zone of indoor positioning in an open area (in this case, in the middle of a hall). For this purpose, several anchors located in not-very-advantageous positions were considered to assess the performance of the positioning in a complex environment. In this case, five anchors were placed in known positions to cover the whole scenario, whereas the tag was placed in different positions of a grid in the middle of the area. In the evaluation scenario, the measurements were taken at each considered position during a period of 60 s, yielding 1500 measurements per position.

The measurement data obtained in both scenarios are available for the scientific community in [9,10], and they can be used to repeat and/or reproduce the experiments described in Section 5. More specifically, the data corresponding to the ML model-generation scenario, which is the same one considered in [5], were published in [9], whereas the data corresponding to the evaluation scenario are available in [10].

## 3. Machine Learning Techniques

In a previous work [5], the authors presented the results obtained by ML algorithms when trained with data corresponding to the ML model-generation scenario, also employing the Pozyx devices described in Section 2.1, to detect NLOS situations between anchors and tags (classification), and also to mitigate the errors obtained by the TWR (regression). The algorithms that exhibited the best results were the k-nearest neighbors (k-NN) for classification and the Gaussian process (GP) for mitigation, which are described in Section 3.1 and Section 3.1.1, respectively. In addition to these algorithms, an MLP (see Section 3.1.2) has also been considered for both classification and mitigation.

Notice that the input features of the considered ML techniques have already been selected in a previous work [18]. More specifically, we chose as features the moving average (with a window of five measurements) of both ranging ($\mu_{\mathrm{ran}}$) and RSS ($\mu_{\mathrm{RSS}}$) obtained from the UWB devices.

### 3.1. k-Nearest Neighbors

k-NN is a classic ML algorithm that can be used in both classification and regression problems [19]. As a classifier, the main idea of this algorithm is to associate each unclassified measurement with the class containing more elements (neighbors) near the measurement. Different metric functions can be used, such as the Euclidean distance, Mahalanobis, City block, or Minkowski. Besides this function, the most important configuration factor of the k-NN is the value of *k* that defines the number of neighbors. Although there are some general recommendations for choosing these parameters, in this work, we opted for using an automatic method to search for the best values in our set. Thus, we considered the Bayesian Optimization method to complete this task. This approach is used to find the optimal values of a so-called *blackbox* function—when there is no information about the function itself beyond its value at some points [20]. Normally, these evaluations have some cost in terms of time or computational effort; hence, the Bayesian Optimization describes a strategy of extracting the maximum information about the function using the lowest number of evaluations. After this process, in our training set, we obtained the values of k=47 and the *City block* distance as the distance function. This distance is defined as:(1)$$d_{1}\left(p,q\right)={∥p-q∥}_{1}=\sum_{i=1}^{n}\left|p_{i}-q_{i}\right|,$$where $d_{1}\left(p,q\right)$ is the *City Block* distance between two vectors p and q of dimension *n*.

#### 3.1.1. Gaussian Process Regression Model

A GP is a generalization of the Gaussian probability distribution in which the distribution does not describe a scalar random variable, but the properties of a function. Based on this idea, it is possible to build regression and classification models with high accuracy and performance [21]. After the training phase, Table 1 shows the hyper-parameters for each mitigator. In this table, *Kernel* indicates the kernel function used and σ indicates the lower bound on the noise standard deviation.

The automatic relevance determination (ARD) Exponential kernel function is defined as:(2)$$k\left(x_{i},x_{j}|\theta \right)=\sigma_{f}^{2}\exp\left(-r\right),$$where $=\sigma_{f}$ is the signal standard deviation and
(3)$$r=\sqrt{\sum_{m=1}^{d}\frac{{\left(x_{im}-x_{jm}\right)}^{2}}{\sigma_{m}^{2}}}.$$

The exponential kernel is defined as:(4)$$k\left(x_{i},x_{j}|\theta \right)=\sigma_{f}^{2}\exp\left(-\frac{r}{\sigma_{l}}\right),$$where $\sigma_{l}$ is the characteristic length scale and *r* is the Euclidean distance between $x_{i}$ and $x_{j}$.

#### 3.1.2. Multilayer Perceptron

An MLP is a classic neural network consisting of an input layer, an output layer, and an arbitrary number of hidden layers. The problem of defining the number of layers and the number of neurons per layer is normally a task that requires testing and measurement of the performance of different configurations until the best one is obtained. Among the different strategies that can be followed, a systematic approach such as Bayesian Optimization is one of the most promising methods [22]. In this work, we used this method to define the architecture of three MLP nets: One to classify the measurements as coming from LOS or NLOS propagation conditions, a second one to mitigate the error in a LOS scenario, and a final one to mitigate the ranging error in a NLOS scenario. The first network outputs a value in the interval [0,1], where 0 indicates a high probability for the measurement to belong to the NLOS set, whereas a value closer to 1 indicates a high probability of belonging to the LOS set. The mitigators output a *ranging* correction for the received measurement that must be added to the original value.

The Bayesian Optimization method generated the hyper-parameters shown in Table 2 for each neural network (NN). All of these networks were trained using the scaled conjugate gradient backpropagation algorithm [23], configured with the parameters shown in Table 3.

## 4. Location Algorithms

Once the ML techniques are applied to the measurements, the filtered and/or corrected ranging data were the input for the algorithm that estimates the tag positions.

For this work, we selected two different types of location algorithms to process the ranging measurements and obtain the position estimates of the UWB device. The first one is based on NLS with a Gauss–Newton iterative procedure to minimize the objective function. The second one is an implementation of the IEKF. Section 4.1 and Section 4.2 describe, respectively, the aforementioned algorithms.

### 4.1. Nonlinear Least Squares with Gauss–Newton

With Pozyx devices, we can obtain an estimation of the TOF of the signal traveling from one emitter to one receiver. Such a time estimate can be easily transformed into a distance estimation simply by multiplying by the speed of light. When we have several distance references from one device (tag) to a set of *L* static devices (anchors) placed at fixed positions around a scenario, we can define:(5)$$r_{\mathrm{TOF},l}=d_{l}+n_{\mathrm{TOF},l}\;l=1,2,\ldots ,L,$$where $r_{\mathrm{TOF},l}$ are the ranging measurements between the tag and the anchor *l*, $d_{l}$ is the actual distance between them, $n_{\mathrm{TOF},l}$ is an error component associated with this measurement and modeled as additive white Gaussian noise (AWGN). Using the Euclidean distance, we have
(6)$$r_{\mathrm{TOF},l}=\sqrt{{\left(x-x_{l}\right)}^{2}+{\left(y-y_{l}\right)}^{2}+{\left(z-z_{l}\right)}^{2}}+n_{\mathrm{TOF},l},\;l=1,2,\ldots ,L,$$where $\left(x,y,z\right)$ are the coordinates of the tag and $\left(x_{l},y_{l},z_{l}\right)$ are the coordinates of each anchor.

Using the NLS approach [24], a position estimate can be obtained starting from (Equation 6) without previously performing any linear approximation like in other approaches [25,26,27]. Thus, we can rewrite (Equation 6) as:(7)$$r_{\mathrm{TOF}}=f_{\mathrm{TOF}}\left(x\right)+n_{\mathrm{TOF}},$$where $f_{\mathrm{TOF}}\left(x\right)$ is a nonlinear function and x is the position to be estimated. Therefore, we can define our cost function as
(8)$$\begin{array}{cc} J_{\mathrm{NLS},\mathrm{TOF}}\left(\tilde{x}\right) & =\sum_{l=1}^{L}{\left(r_{\mathrm{TOF},l}-\sqrt{{\left(\tilde{x}-x_{l}\right)}^{2}+{\left(\tilde{y}-y_{l}\right)}^{2}+{\left(\tilde{z}-z_{l}\right)}^{2}}\right)}^{2}\\ \; & ={\left(r_{\mathrm{TOF}}-f_{\mathrm{TOF}}\left(\tilde{x}\right)\right)}^{T}\left(r_{\mathrm{TOF}}-f_{\mathrm{TOF}}\left(\tilde{x}\right)\right), \end{array}$$where $\tilde{x}=\left[\tilde{x}\;\tilde{y}\;\tilde{z}\right]$ is the optimization variable for x. Using a least squares method, we have that our best estimation is

(9)$$\hat{x}=\arg\min_{\tilde{x}}J_{\mathrm{NLS},\mathrm{TOF}}\left(\tilde{x}\right).$$

Finding this minimum point is not trivial and many different techniques can be applied, such as [28] or [29]. In this work, we chose an iterative Gauss–Newton method that approximates the solution in each iteration. Thus, the equation that defines this evolution is
(10)$$\begin{array}{c} {\hat{x}}^{m+1}={\hat{x}}^{m}+{\left(G^{T}\left(f_{\mathrm{TOF}}\left({\hat{x}}^{m}\right)\right)\right)}^{-1}G^{T}\left(f_{\mathrm{TOF}}\left({\hat{x}}^{m}\right)\right)\left(r_{\mathrm{TOF}}-f_{\mathrm{TOF}}\left({\hat{x}}^{m}\right)\right), \end{array}$$where G is the Jacobian matrix of $f_{\mathrm{TOF}}\left({\hat{x}}^{m}\right)$ calculated at ${\hat{x}}^{m}$.

### 4.2. Iterative Extended Kalman Filter

The Kalman filter is a well-known algorithm to estimate the hidden state of a system given some observation variables, and is widely applied to positioning problems. The original Kalman algorithm provides an exact solution for this estimation problem in systems where the observations are in a linear state together with Gaussian-distributed noise sources. However, when some of these assumptions do not hold, numerous variations were proposed to overcome these limitations, such as the Extended Kalman filter [30], the Unscented Kalman filter [31], and particle filters [32].

In this work, the ranging observations are nonlinear on the tag and anchor positions; hence, we implemented an IEKF, which linearly approximates the actual observation functions and solves the associated maximum a posteriori probability problem using as prior information a prediction on the state when a new observation arrives. In particular, the state $x\in R^{6}$ contains the position and velocity on the three axes, with a prediction model
(11)$$\begin{array}{cc} x_{t|t-1} & =Gx_{t-1}+m_{t}, \end{array}$$
(12)$$\begin{array}{cc} P_{t|t-1} & =GP_{t}G_{t}^{H}+C_{m}, \end{array}$$where G=[I,ΔtI;0,I], $m_{t}∼N\left(0,C_{m}\right)$ is a noise component caused by the unpredicted acceleration, $P_{t}$ is the prediction covariance with $C_{m}=\left[\Delta t^{4}/4I,\Delta t^{3}/2I;\Delta t^{3}/2I,\Delta t^{2}I\right]$, and Δt is the time delay since the last received observation. The observation model is
(13)$$\begin{array}{c} y_{t}=r\left(x_{t}\right)+n_{t}, \end{array}$$where r(·) is the ranging function for all the anchors and $n_{t}∼N\left(0,C_{n}\right)$ is the observation error component, with $C_{n}$ being a diagonal matrix with the estimated error variance of each anchor on its main diagonal. The IEKF iteratively searches for the solution of the state for this observation model with prior information from (11) and (12). The function r(·) is linearized using the vectors
(14)$$\begin{array}{c} r_{l}={\left[\left(x-x_{l}\right)/d_{l},\left(y-y_{l}\right)/d_{l},\left(z-z_{l}\right)/d_{l},0\right]}^{T}, \end{array}$$obtaining the Jacobian matrix $R\left(x\right)=\frac{\partial r(x)}{\partial x}={\left[r_{1},\ldots ,r_{L}\right]}^{T}$, where x,y, and *z* are the state variables corresponding to the position of the tag. Hence, state estimations are iteratively updated for each group of received rangings at the time instant *t* as
(15)$$\begin{array}{c} x_{i}=x_{t|t-1}+P_{t|t-1}R_{i}^{T}{\left(R_{i}P_{t|t-1}R_{i}^{T}+C_{n}\right)}^{-1}\times \left(y_{t}-r\left(x_{i-1}\right)-R_{i}\left(x_{t|t-1}-x_{i-1}\right)\right),i=1,\ldots ,I, \end{array}$$where $x_{0}=x_{t|t-1}$ and the matrix $R_{i}=R\left(x_{i-1}\right)$ is updated after each iteration. More details can be found in [30].

## 5. Experiments and Results

### 5.1. Results Obtained Considering Different Scenarios for Generating and Evaluating the Models

In this section, we consider, on the one hand, the classification and mitigation algorithms obtained through the training, validating, and testing carried out using the data recorded in the ML model-generation scenario (see Figure 2a). On the other hand, the data extracted from the measurements in the evaluation scenario shown in Figure 2b were employed to assess the performance of the aforementioned algorithms. More specifically, a tag was moved between different points of the grid in Figure 2b to simulate the movement of a real target. Notice that this evaluation process ensures that the generated models are employed in a scenario different from the one used for their generation, leading to the most general and realistic situation possible.

Figure 3 shows RSS versus ranging values measured in the ML model-generation scenario and corresponding to LOS (blue) or NLOS (yellow) propagation conditions. Figure 3 also plots the values measured in the evaluation scenario after being classified as LOS (green) or NLOS (magenta) by the NN classifier. The superposition of the values from the two scenarios demonstrates that (1) the NN is able to distinguish correctly between LOS and NLOS, even though both scenarios are different, and (2) the relationship between RSS and ranging values is significantly different in both scenarios, especially in the case of NLOS.

In order to get a clear idea of the effect of each element on the final result, positioning error results were obtained using different combinations of the following algorithms:The k-NN algorithm as the classifier and the GP regression model as the mitigator.The NN as both classifier and mitigator.The IEKF as the positioning algorithm.The NLS with Gauss as the positioning algorithm.

In addition, for each of the considered combinations, different configurations of the positioning algorithms were established:**No ignore.** This is the configuration that serves as the reference for the location error. In this case, the positioning algorithms employ all raw measurement data recorded in the scenario, without any previous classification or mitigation processes. That is to say, the measurement data are the direct input to the positioning algorithms.**Ignore NLOS and no mitigation.** Either k-NN or NN is used to classify each measured value as LOS or NLOS. Subsequently, the positioning algorithms will only be fed with the measurement data in the LOS category, without any mitigation, and ignoring data classified as NLOS.**Ignore NLOS with at least four anchors and no mitigation.** NLOS data are classified and ignored as in configuration 2, but while ensuring at least four ranging values for the positioning algorithms, enabling algorithms such as the NLS, which otherwise could not provide an estimate in the three dimensions. When less than four values are classified as LOS in one iteration, then the necessary values classified as NLOS are included. More specifically, the ranging values classified as NLOS with the lowest score, i.e., those which are less likely to belong to the NLOS category, are selected to replace the missing LOS values. Again, no mitigation mechanism is applied to the ranging data. This configuration was not tested with the IEKF, as this algorithm does not need to have at least four values to estimate a position.**Ignore NLOS with mitigation.** NLOS ranging values are classified and removed from the positioning process. The errors in the remaining ranging values (classified as LOS) are mitigated with one of the algorithms before being sent to the positioning algorithms to finally produce position estimates.**Ignore NLOS with at least four anchors and mitigation.** This is a combination of configurations 3 and 4: NLOS ranging values are ignored, but the ones more likely to be classified as LOS are considered to ensure four ranging values in each iteration. Before running the positioning algorithm, each measurement value is passed through an LOS mitigator. Again, this configuration was not tested with the IEKF, as this algorithm does not need to have at least four values to estimate a position.**No ignore with mitigation.** The same as configuration 1, but each measurement is passed through a mitigator—modeled according to the category determined by the classifier (i.e., LOS or NLOS)—before being processed by the positioning algorithms.

Figure 4a shows the empirical cumulative distribution function (ECDF) for the location error obtained in the evaluation scenario with the IEKF algorithm, using the k-NN classifier and two GP-based models (LOS and NLOS) for mitigation. The results in Figure 4a show that, when using all of the raw measured values (configuration 1) to estimate the positions (no classification or mitigation), 90% of the errors are below 0.83 m. In this case, with a reduced number of anchors and many ranging values obtained under NLOS propagation conditions, the error is very large compared to what is typically obtained with UWB technology when there is a good LOS. Obviously, this situation is due to the presence of several anchors simultaneously under NLOS propagation conditions. Although this situation could be improved considerably by placing these tags in more convenient positions, we opted for a more complex and realistic situation, since it is not always possible to place them in the best location. The margin of improvement achieved by the considered algorithms and configurations is assessed under this disadvantageous evaluation scenario.

Figure 4a shows that mitigating the measurements once they were classified (magenta line labeled as 4) does not reduce the error and even worsens it. This is because the relationship between the ranging and RSS values in the ML model-generation scenario is significantly different from that in the evaluation scenario, especially for the NLOS case, since it is very dependent on the geometry of the environment (see Figure 3). Therefore, the model under consideration attempts to correct the data in an erroneous manner, since it is trained to conform to the values obtained in the ML model-generation scenario and exhibits difficulties when operating on the different values from the evaluation scenario. Additionally, wrong classifications lead to the application of NLOS mitigation models to LOS data and vice versa, also contributing to increase in the error.

The remaining curves in Figure 4a show the effect of discarding ranging values classified as NLOS. The best result is obtained when no mitigation is applied (line labeled as 2) and all the measurements classified as NLOS are ignored by the location algorithm. On the other hand, when the mitigator (line labeled as 3) is applied, even if it is only on ranging values classified as LOS, there is a small performance degradation.

According to the results in Figure 4a, ignoring NLOS without mitigation (configuration 2) leads to a significant improvement, in which 90% of the errors are below 0.25 m compared to 0.83 m when NLOS ranging values are not excluded (black line). This confirms that the classifier trained, validated, and tested with measurements from the ML model-generation scenario was generic enough to be used in another different scenario and produces satisfactory results.

Figure 4b shows the ECDF using IEKF, but this time employing the NN for both classification and mitigation. Again, the best result is also achieved when the ranging values are classified and the NLOS ones are discarded (configuration 2). The result in this configuration is almost the same as with the k-NN (see Figure 4a), reaching 0.19 m of error for 90% of the estimates, compared to 0.21 m obtained when the k-NN is used (see Figure 4a). When the mitigation is added (orange curve, labeled as 3), again, there is some performance degradation, yielding values lower than those produced with k-NN and GP mitigators. This is because NN—despite obtaining better results in mitigation during training—is more sensitive to changes in the environment during the evaluation due to certain overfitting associated with it.

To analyze the effect of a different positioning algorithm, Figure 4c,d show the ECDF using the NLS-based algorithm to estimate the positions while considering k-NN/GP and NN, respectively. The main differences between these figures and the previous ones using IEKF are in configurations 2 (solid green) and 4 (solid orange), which correspond to the cases in which NLOS ranging values are ignored (and mitigated in the case of configuration 4).

Figure 4c,d show that with these configurations, the error increases with respect to the reference configuration (without ignoring NLOS or applying mitigation). This behavior is due to the way that the NLS-based algorithm works. This type of algorithm, like others that try to calculate a position by directly using the trilateration equations, cannot generate a new estimation if they do not have a minimum number of ranging values, hence requiring at least four values in the case of a 3D positioning. Thus, when NLOS ranging values are ignored, it can happen that in several iterations, the algorithm does not have that minimum number of ranging values, so a new position is not generated and the previous one is maintained.

However, when at least four ranging values are available (dashed curves in Figure 4c,d), although some of them have been classified as NLOS, the results (dashed curves) again improve the error records obtained in the reference case (black curve, configuration 1). As with the IEKF, such improvement is more pronounced when mitigation is not applied and only the NLOS measurements are ignored (green dashed curve, configuration 3). We can also see how with this configuration, the results are almost the same in both cases, using NN or k-NN as classifiers. This means that the two sets (LOS and NLOS) can be clearly separated. In the case of configuration 5 (mitigation after ignoring NLOS), the solution based on k-NN seems to give the best results for most parts of the measurements, but there are outliers that introduce a large error.

Figure 5a,b show a comparison of the different configurations from the point of view of the localization mean absolute error (MAE). The figure shows the values together with their 95% confidence interval.

Figure 5a shows the results when k-NN is used for classification and GP for mitigation. The first configurations (1 and 2) correspond to the cases in which the location is carried out using all available measurements, without applying any type of filter or mitigation. It can be seen that the values corresponding to IEKF (label 1, gray) and NLS (label 2, crossed out, gray) are practically the same, around 0.38 m. These configurations are the basis of the comparison, as they mark the expected error before applying the techniques proposed in this work.

Configurations 3, 4, and 5—shown in Figure 5a—are those in which a classification process is applied to identify the NLOS samples and remove them from the set used by the localization algorithms. When the IEKF is used (label 3, dark green), the error value decreases significantly, falling below 10 cm (0.084 m). However, when the NLS is employed, the error increases above 75 cm because, when NLOS values are eliminated, the minimum number of samples required by this algorithm in each iteration is not guaranteed and, when this occurs, the algorithm is not able to generate a new position, and the previous value is maintained. If this situation holds through several consecutive iterations, the error grows quickly. In configuration 5 (crossed out light green), the rule is applied to always maintain at least 4 *range* measurements for each iteration, *filling in* with NLOS measurements in case there are not enough LOS values to reach that number. With this strategy, the MAE value falls below that of the original error value. Obviously, the final value of (0.155
m) is higher than that obtained with the IEKF (0.084
m) because the NLS algorithm is forced to work with a small percentage of NLOS measurements.

The configurations 6 to 8—shown in Figure 5a—correspond to the cases in which mitigation is applied. Thus, the values labeled as 6 and 7 correspond to the cases where, after ignoring the measurements classified as NLOS, a mitigation process is applied over the remaining ones. For the IEKF case (label 6, orange), the error value is 0.136
m, which is better than the original one of 0.38
m, but it does improve the 0.084
m of configuration 3, in which no mitigation was applied. As mentioned above, this is due to the differences in the relationship between ranging and RSS values found in the ML model-generation and evaluation scenarios (see Figure 3). The configuration labeled as 7 (crossed out, orange) for the NLS increases the error due to the same reasons as those explained for configuration 4. The approximation in which this minimum number is maintained (at the cost of introducing some NLOS measurements) corresponds to configuration 8 (crossed out, red). In this case, the MAE improves the original value, but less than in the IEKF case (0.330
m instead of 0.136
m).

Finally, the configurations 9 and 10—shown in Figure 5a—show the results of when NLOS values are included and mitigation is applied independently to LOS and NLOS measured values. The results are identical for both localization algorithms and are even worse than the error obtained in configurations 1 and 2, when no mitigation was applied (about 5 cm of worsening in the MAE). Once again, this result demonstrates that the mitigation strategy is strongly dependent on the ML model-generation scenario; hence, in a significantly different evaluation scenario, it is not able to improve the MAE.

Figure 5b shows the results corresponding to the previous configurations, but when NN is used as a classifier and mitigator. The values obtained are very similar to those provided by the k-NN and GP, especially when only the classification is applied to ignore NLOS measurements, yielding almost the same results. The main difference appears in the configurations where mitigation is applied. In this case, the version with IEKF performs slightly worse, whereas the version with NLS and at least four values performs slightly better. This indicates that LOS mitigation is better with GP (due to a lower fitting with the training set data), whereas classification is better with NN. This means that the NLS version with at least 4 values includes NLOS values in smaller proportions, and that those added values are sometimes values close to being classified as LOS.

Given these results, we can conclude that, in a scenario in which the propagation condition (LOS or NLOS) is unknown, the detection of NLOS ranging values is feasible using ML models that have been trained, validated, and tested with measurements from a different scenario in which we know beforehand, for a given measured value, whether the propagation is LOS or NLOS. This allows for reuse of the proposed ML classifiers in scenarios different from those considered for the training. However, the ML mitigation model considered in this work is strongly dependent on the training scenario; hence, reusing it in a significantly different scenario is counterproductive.

### 5.2. Results Obtained for Model Generation and Evaluation Based on the Same Scenario

Section 5.1 compared the location results obtained in a given evaluation scenario by applying a series of ML models to filter and mitigate measurements affected by NLOS. These models were trained with data from a measurement campaign carried out in a different scenario, the ML model-generation scenario. Because of the differences between both scenarios, the models were not able to improve the baseline error values in the case of mitigation. To confirm this assumption, a new study is presented below that replicates the experiments described in Section 5, but in a new scenario. This scenario differs from the previously used evaluation scenario (see Figure 2b) in that the measurements obtained in it are similar to those obtained in the ML model-generation scenario (see Figure 2a).

To do this, since it would be impossible to replicate a configuration of beacons in the ML model-generation scenario in the exactly same way as in the evaluation scenario, it was decided to carry out a simulation-based approach. This simulation is based on the UWB simulator described in [8] and developed by the authors. The simulator is built on the Gazebo simulation platform [33], a multi-platform software consisting of several components and focused on the virtual simulation of real physical environments. Five UWB anchors were placed in the same positions as in the evaluation scenario described in Section 2.2, shown in Figure 2b. The final model of the scenario is drawn in Figure 6.

The reason for why this simulator can be used as an approximation to the problem is that measurements from the ML model-generation scenario shown in Figure 2a and considered in Section 5.1 were employed to recreate the evaluation scenario shown in Figure 2b. More specifically, the construction of the simulator was based on the following steps:A measurement campaign was carried out in the same real-world ML model-generation scenario considered in this work (see Figure 2a).These measurements were used to train, validate, and test a series of ML models capable of providing estimates of ranging, ranging variance, RSS, and RSS variance from two input values: The distance between the tag and the simulated anchor, and the propagation conditions between them: LOS, NLOS-Soft, or NLOS-Hard.Finally, a 3D ray-tracing model was developed to estimate the type of scenario between the tag and the anchors. More specifically, the simulator assigns the LOS type when the signal propagates from the tag to the anchor without touching any obstacles. It assigns NLOS-*Soft* when the signal encounters an obstacle between them, but is able to cross it (according to some configuration parameters of the simulator). Finally, the simulator assigns the NLOS-Hard type in the case of finding a big obstacle between the tag and the anchors, but it is capable of tracing a connection between both after bouncing off of a wall or the floor of the simulated scenario. Notice that, for the sake of simplicity, rebounds on the ceiling are not considered in the current version of the simulator.

Once the simulated scenario was built, ranging and RSS values were simulated for the same points in which measurements were taken in the real-world evaluation scenario. Thus, the simulated UWB tag was placed on the points of a 3×3 grid, and values were simulated during a period of 60 s at each point, the same capture time as in the real case. After obtaining the simulated values, the same process as in the case with measurements was followed: NLOS classifiers and mitigators (based on k-NN, GP, and NN) were used to filter the simulated data; and finally, the results fed the localization algorithms (the one based on NLS and the IEKF). The same configurations described in Section 5.1 were evaluated.

Figure 7 shows the error values obtained for each of the analyzed configurations (the same that can be seen in Figure 4 for the measurements) when using the IEKF and the NLS. In the same way, Figure 8 shows the MAE values for such configurations.

Figure 7 and Figure 8 reveal that, in general, the values obtained by the simulator led to error values lower than those obtained from the real measurements and shown in Figure 4 and Figure 5. A reference case is the *no ignore* configuration, since it analyzes the positioning error without considering any previous ML technique. As detailed in Table 4 and Table 5, the MAE difference between the measured and simulated environment is practically the same for both location algorithms (IEKF and NLS), i.e., about 15 cm.

It is also important to check the data in the other combinations, as they give us a hint about how similar the NLOS set of values obtained in both cases is. Thus, for the IEKF case, it can be seen how the differences in the MAE with respect to the data obtained from the measurements are quite small, always below 13 cm and falling below 2 cm for the configurations in which the measurements classified as NLOS are ignored (both with k-NN and with NN). Logically, in the case of the simulated environment, the results are a little better, as they consider similar evaluation and ML model-generation environments. This corroborates the results shown in Section 5.1, where the good performance of the LOS/NLOS classification together with its benefit in the positioning algorithm were shown.

An important difference is that, for the simulated case, the configuration that applied mitigation after ignoring the NLOS values obtains a better result than the configuration in which only the NLOS values are eliminated and no mitigation is carried out. As already explained in the analysis of the measurement results in Section 5.1, this behavior was expected when the training set covered the entire sampling space of the evaluation scenario. In the real measurements, there were certain values in the evaluation scenario that were not represented in the ML model-generation set (see Figure 3) since they were obtained in a different scenario; hence, the mitigation did not improve the results. With the simulation, this is different, since the simulated values came from a model based on the ranging and RSS data obtained in the ML model-generation scenario. Therefore, in the simulation, both the classifiers and the mitigators were trained with similar data; hence, their performance was expected to be much better, as reflected by the MAE results.

With respect to the MAE results obtained by using the NLS algorithm (see Table 5), the main differences are found in the configurations where the NLOS measurements were eliminated and mitigation was applied or not, but without reserving a minimum number of 4 anchors in each iteration of the algorithm. In the real measurements case, these configurations produced a very high error level—over 75 cm for MAE—mainly because too many samples were eliminated, and the algorithm could not generate new positions. In the simulation, although the same relation was maintained with respect to the version that reserves a minimum of 4 anchors (the latter reduces the MAE), the value of the MAE for the simulation is much lower (staying around 16 cm). This is explained in part by the better efficiency of the classifier in the simulation, which lead to the elimination of fewer erroneous samples; hence, the algorithm remains useful for few occasions without the minimum number of values required to perform the calculations.

## 6. Conclusions

In this work, we have analyzed the performance of a complete low-cost UWB positioning system based on TWR considering ML algorithms that are capable of detecting and mitigating UWB ranging measurements from an NLOS environment. Unlike other similar works, our approach assessed the performance of these algorithms when the training, validating, and testing phases were carried out with measurements from a given scenario, but the evaluation was performed with data from a different one. This made it possible to determine the mitigation and classification models after a first phase carried out in a controlled environment, which allowed us to characterize LOS and NLOS situations. In a later stage, the obtained models were applied to a new evaluation scenario.

We have tried six different configurations, including the four different algorithms considered in this work: k-NN as the classifier and GP regression model as the mitigator, NN for both classification and mitigation, and two positioning algorithms: IEKF and NLS with Gauss. The obtained results revealed that using the IEKF as the positioning algorithm and the k-NN or NN for detecting NLOS ranging values leads to the best performance in the evaluation scenario, exhibiting an MAE of only 0.084–0.085 m, in comparison with an MAE of 0.38 m obtained when no detection or mitigation at all were performed. However, the results showed that this improvement in performance is due to the incorporation of the classification process, but that the mitigation of NLOS effects did not work properly, as they are strongly dependent on the environment in which the algorithms were trained.

To compare the performance of the proposed solution when using models trained and evaluated in the same scenario versus using models trained and evaluated in different ones, a simulation was carried out. For this purpose, a UWB simulator was used and a virtual scenario copied from the evaluation scenario was designed. The results showed that when the models are generated and evaluated with measurements from the same scenario, the application of mitigation strategies improves the original MAE values, something that did not occur when the models were generated and evaluated with data captured in different scenarios.

## Figures and Tables

**Figure 1 sensors-19-05438-f001:**
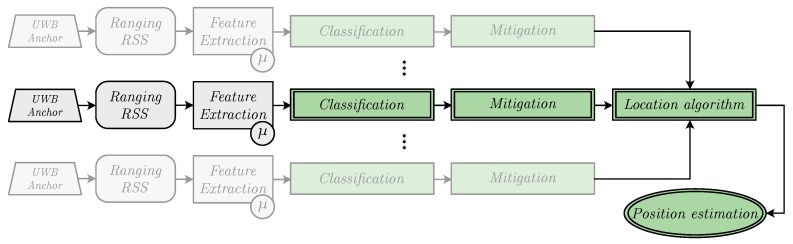
Block diagram showing the pipeline of the whole localization process.

**Figure 2 sensors-19-05438-f002:**
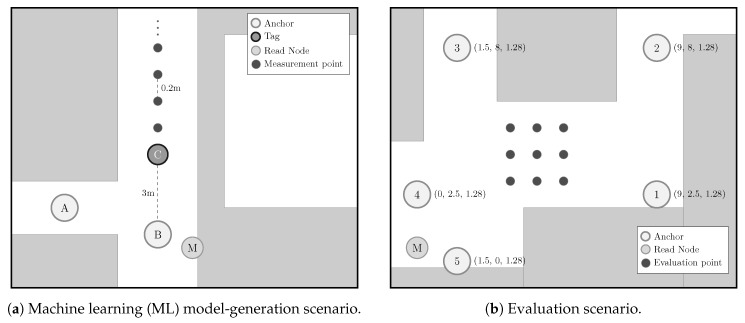
(**a**) Scenario previously considered in [5] to train, validate, and test all of the ML-based models which are evaluated in this work; the new scenario is shown in (**b**).

**Figure 3 sensors-19-05438-f003:**
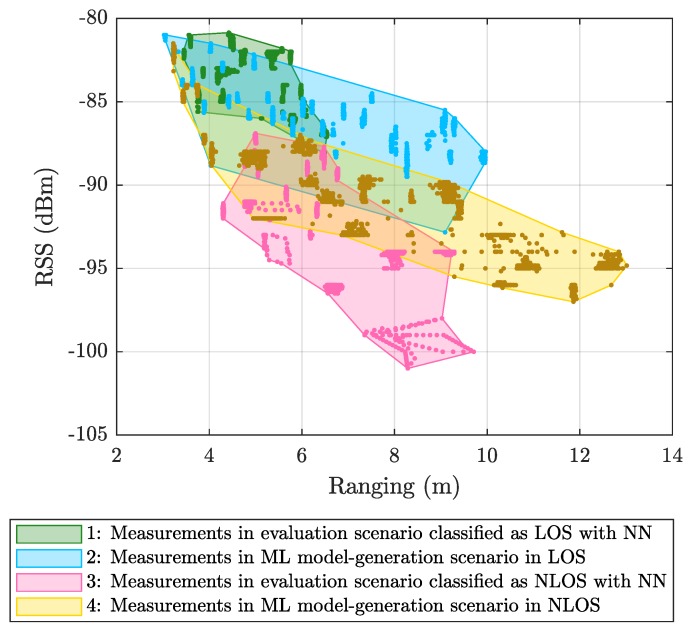
Classification results in the evaluation scenario using NN compared to the line-of-sight (LOS) and non-line-of-sight (NLOS) values measured in the ML model-generation scenario.

**Figure 4 sensors-19-05438-f004:**
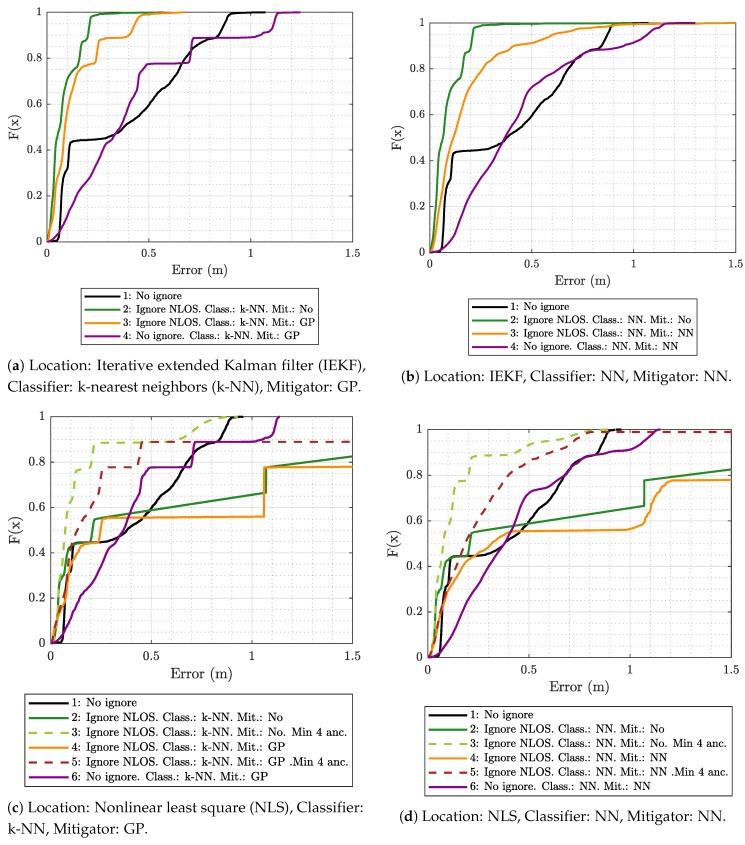
Empirical cumulative distribution function (ECDF) of the location error in the evaluation scenario using different combinations of location algorithms, classifiers, and mitigators.

**Figure 5 sensors-19-05438-f005:**
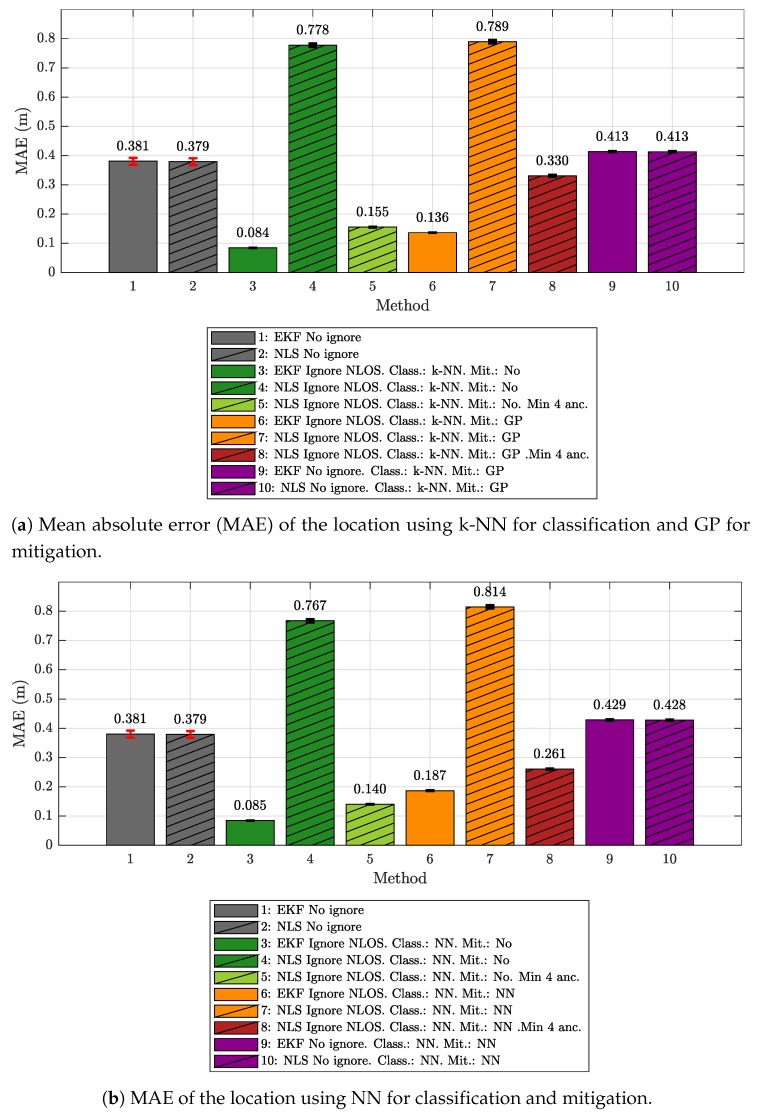
MAE of the location.

**Figure 6 sensors-19-05438-f006:**
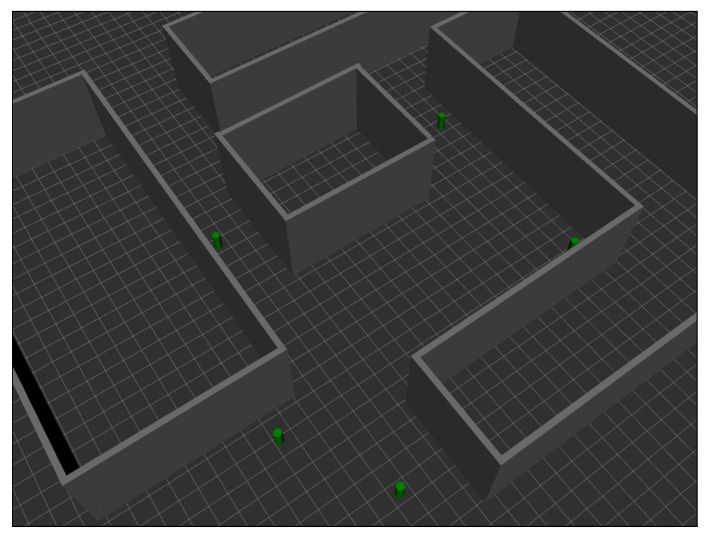
Grid simulation scenario.

**Figure 7 sensors-19-05438-f007:**
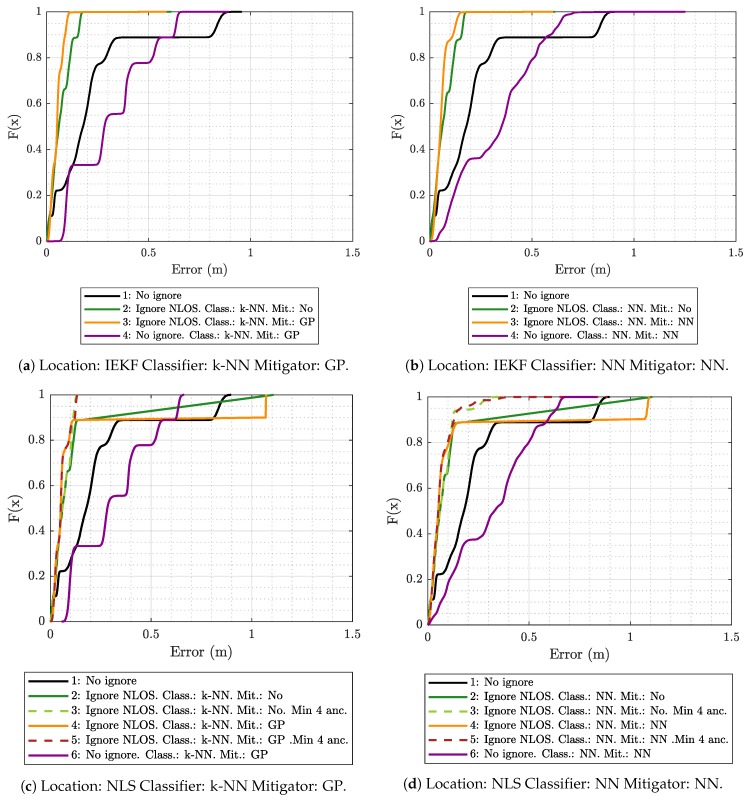
ECDF of the location error using different combinations of location algorithms, classifiers, and mitigators in the simulation.

**Figure 8 sensors-19-05438-f008:**
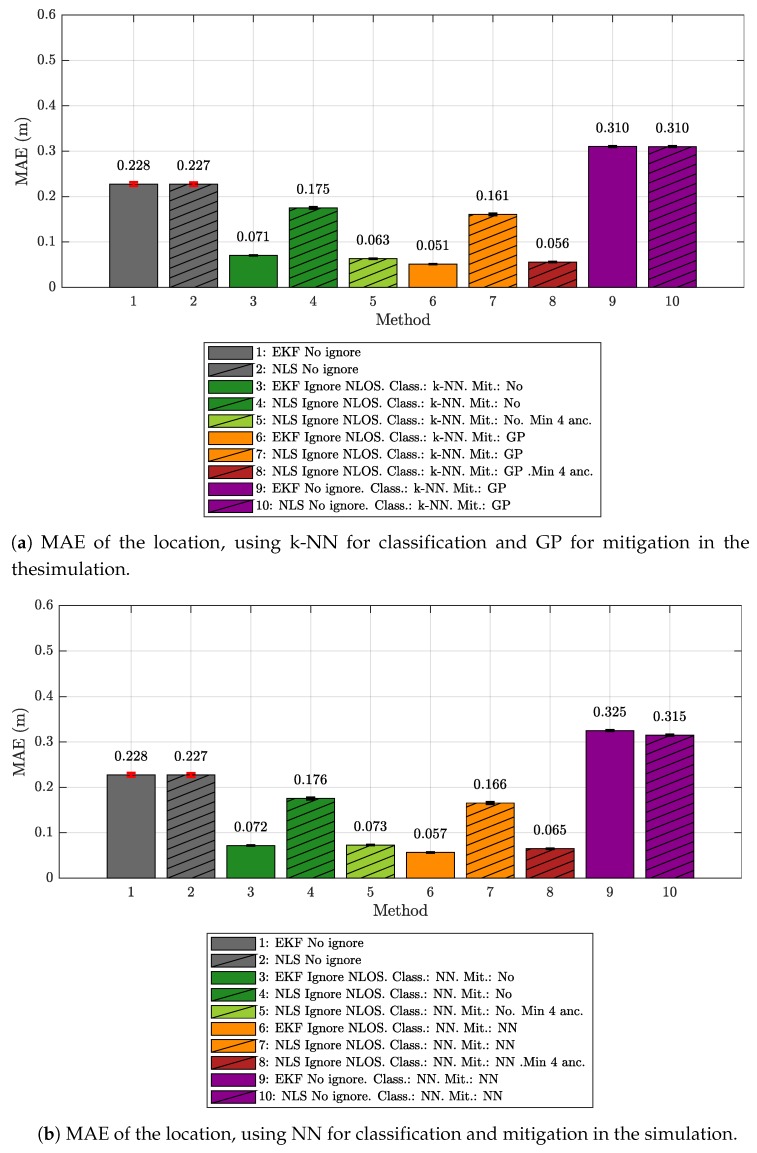
MAE of the location in simulation.

**Table 1 sensors-19-05438-t001:** Hyper-parameters of Gaussian process (GP) mitigators.

Mitigation Class	Kernel	σ
LOS	ARD Exponential	40.43
NLOS	Exponential	49.82

**Table 2 sensors-19-05438-t002:** Hyper-parameters of the NN.

NN	# Hidden Layers	# Neurons	Act. Fun. Hidden Layers	Act. Fun. Last Layer
Classification	3	$\left[46,14,24\right]$	Hyperbolic tangent sigmoid	Soft max
Mitigation LOS	2	$\left[21,11\right]$	Hyperbolic tangent sigmoid	Linear
Mitigation NLOS	2	$\left[65,69\right]$	Hyperbolic tangent sigmoid	Linear

**Table 3 sensors-19-05438-t003:** Scaled conjugate gradient backpropagation parameters.

Parameter	Value
Maximum Epochs	1000
Maximum Training Time	∞
Performance Goal	0
Minimum Gradient	1 × 10^−6^
Maximum Validation Checks	6
σ	5 × 10^−5^
λ	5 × 10^−7^

**Table 4 sensors-19-05438-t004:** Measured versus simulated MAE comparison using IEKF.

IEKF	Measured	Simulated	Difference
No Ignore	0.381	0.228	0.153
Ignore NLOS. Class.: k-NN. Mit.: No.	0.084	0.071	0.013
Ignore NLOS. Class.: k-NN. Mit.: GP.	0.136	0.051	0.085
No Ignore. Class.: k-NN. Mit.: GP.	0.413	0.310	0.103
Ignore NLOS. Class.: NN. Mit.: No.	0.085	0.072	0.013
Ignore NLOS. Class.: NN. Mit.: NN.	0.187	0.057	0.130
No Ignore. Class.: NN. Mit.: NN.	0.429	0.325	0.124

**Table 5 sensors-19-05438-t005:** Measured versus simulated MAE comparison using NLS.

NLS	Measured	Simulated	Difference
No Ignore	0.379	0.227	0.152
Ignore NLOS. Class.: k-NN Mit.: No.	0.778	0.175	0.603
Ignore NLOS. Class.: k-NN Mit.: No. Min 4.	0.155	0.063	0.092
Ignore NLOS. Class.: k-NN Mit.: GP.	0.789	0.161	0.628
Ignore NLOS. Class.: k-NN Mit.: GP. Min 4.	0.330	0.056	0.2
Ignore NLOS. Class.: NN Mit.: No.	0.767	0.176	0.591
Ignore NLOS. Class.: NN Mit.: No. Min 4.	0.140	0.073	0.067
Ignore NLOS. Class.: NN Mit.: NN.	0.814	0.166	0.648
Ignore NLOS. Class.: NN Mit.: NN. Min 4.	0.261	0.065	0.196
No Ignore. Class.: NN Mit.: NN.	0.428	0.315	0.113

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
