# Peer review of "Environmental Cross-Validation of NLOS Machine Learning Classification/Mitigation with Low-Cost UWB Positioning Systems"

_sensors, 2019, doi:10.3390/s19245438_

Round 1

Reviewer 1 Report

I would like to thank the authors for better explaining the novelty of their work and how it improves their previous work published in Sensors. In fact, this part was misunderstood, due to some misleading description in the Introduction.

I believe that the new version is significantly improved and the new Section 5.2 (added according to my previous comment) gives some interesting insights on the localization system performance. Therefore, I do not have further comments and I believe that publication can be now pursued.

Reviewer 2 Report

Overall good article. Small improvements can be mad:

Lines 65 to 67: MLP are part of the ML method. Here the description is unclear / very unfavourable.

Lines 73 and 113 ff: Training and test data should be created under the same / similar conditions so that the performance of the procedure can be evaluated. Separation of errors of the procedure and errors of the scenario is no longer possible. Better use external and more accurate reference positions. In any case, the procedure must be justified and explained in more detail.

Lines 115 et seq. and Figure 2: Designation of anchors inconsistent in 2a and 2b. Why A, B, M and then 1 -5. Why different measurement times? Why different number of anchors? Many parameters are changed for training and testing, which makes it impossible to interpret the results later during fundamental investigations. How often are the measurements repeated per scenario?

Line 142 f: The information that experiences with the same scenarios already exist should be given in the introduction and a explanation for the described measurements should be provided.

Table 1: No information on selected functions. ARD exponential and exponential must be explained, units of sigma missing and too many decimal places.

Line 160 ff: It is not clear which data / values are used in the input layer. What are the output values of the individual artificial neural networks (KNN)? For the classification probably two classes (measurement wrong and measurement right?). For the reduction of the error a parameter (?), but then the choice of function of the output layer should be explained. How are the KNNs chained? Why not use two large networks to mitigate the error after LOS and NLOS? Why not just one KNN that doesn't distinguish between NLOS and LOS? No explanation for final network architecture.

P has not been declared for Formula 9.(line 188)

Table 3: What is meant by Maximum Training Time = Infinite. Are there other abort criteria besides the 1000 epochs (Should it be)? Which metrics were observed during training or should be optimized? Which performance was determined for the classification? Which performance was determined for the reduction of the error?

And a more basic question based on Figure 3: If you see the result in Figure 3, the question arises whether you need neural networks for this, or whether you can also establish a direct functional correlation. What is the advantage of this method?

Author Response

This manuscript is a resubmission of an earlier submission. The following is a list of the peer review reports and author responses from that submission.

Round 1

Reviewer 1 Report

A good quality paper about the UWB Positioning Systems is presented. The text is fairly well written and the paper is clearly structured. Authors are recommended to add a comparison table, before the Conclusion, to compare the fundamental parameters of the employed algorithms in the different configurations.

Reviewer 2 Report

This paper does not describe any advantages and improvement scheme of mitigator, but simply applies it roughly. Since mitigator fail to work without modification, you should not mention it in great detail. Maybe you should try something else, like multilayer neutron networks. The method proposed in this paper is not in any comparison with the contributions made by others. UWB technology is very mature today, so you should compare your contribution to your peers, not just a few of your own scenarios. The text is too tedious. For example, there is little difference between the experimental results of IEKF method and NLS method, maybe you should describe them together. Many references need to be re-annotated. For example, "…with tree layers and [46, 14, 24] neurons…", in lines of 158 , compared to your 30 references.

Reviewer 3 Report

In this paper, the authors extend their previous work on machine learning (ML)-based identification and mitigation of non-line-of-sight (NLOS) communication links. They consider a similar approach, by extending the analysis to other ML algorithms and the use of different sets for training and testing.

Even if the paper is relatively well presented, I honestly believe that its novelty is very limited. In fact, extending a previous (journal) work testing with other algorithms and considering the robustness against changes in the used dataset is not sufficient to justify another journal paper. This may be worthy of at most a conference publication.

Moreover, note that you want to extend the results to a scenario where one dataset is used for training and the other for testing. However, I would like to see a performance comparison with respect to the previous scenario where training and testing were performed using the same dataset? What is the performance loss? How robust is the system?